# Foliar Pectins and Physiology of Diploid and Autotetraploid Mango Genotypes under Water Stress

**DOI:** 10.3390/plants12213738

**Published:** 2023-10-31

**Authors:** Andrés Fonollá, José I. Hormaza, Juan M. Losada

**Affiliations:** Institute for Mediterranean and Subtropical Horticulture ‘La Mayora’ (IHSM La Mayora—CSIC—UMA), Avda. Dr. Wienberg s/n, 29750 Malaga, Spain; afonolla@gmail.com (A.F.); ihormaza@eelm.csic.es (J.I.H.)

**Keywords:** autotetraploids, drought, leaves, *Mangifera indica*, pectins, stomatal conductance, water potential

## Abstract

The cultivation of mango in Mediterranean-type climates is challenged by the depletion of freshwater. Polyploids are alternative genotypes with potential greater water use efficiency, but field evaluations of the anatomy and physiology of conspecific adult polyploid trees under water stress remain poorly explored. We combined field anatomical evaluations with measurements of leaf water potential (Ψ_l_) and stomatal conductance (G_s_) comparing one diploid and one autotetraploid tree per treatment with and without irrigation during dry summers (when fruits develop). Autotetraploid leaves displayed lower Ψ_l_ and G_s_ in both treatments, but the lack of irrigation only affected G_s_. Foliar cells of the adaxial epidermis and the spongy mesophyll contained linear pectin epitopes, whereas branched pectins were localized in the abaxial epidermis, the chloroplast membrane, and the sieve tube elements of the phloem. Cell and fruit organ size was larger in autotetraploid than in diploid mango trees, but the sugar content in the fruits was similar between both cytotypes. Specific cell wall hygroscopic pectins correlate with more stable Ψ_l_ of autotetraploid leaves under soil water shortage, keeping lower G_s_ compared with diploids. These preliminary results point to diploids as more susceptible to water deficits than tetraploids.

## 1. Introduction

Mango (*Mangifera indica*, Anacardiaceae) ranks fifth in terms of production among perennial fruit crops worldwide and is a staple crop for food security in many countries with tropical climates [1], such as India or Brazil; its cultivation is increasing in regions with subtropical and Mediterranean climates [2]. While the Mediterranean basin of the south of Spain is the farthest region from the equator with a significant commercial mango production, climate change predictions point to extended periods of drought that may drive irrigation shortages, likely jeopardizing productivity. Mango trees are known for their tolerance to soil water scarcity [3], but in Mediterranean climates irrigation is mandatory during the summer months, when fruit development takes place. Yet, the physiological changes in mango related to the lack of irrigation during fruit development are not well understood.

While most commercial mango cultivars are diploid, some polyploid genotypes are available, but scarce. Those polyploid genotypes have shown a higher ability to retain water in their leaves compared with diploids [4,5]. Recent studies have revealed that autotetraploid mango seedlings of some varieties show an isohydric strategy combined with higher proline levels in their leaves under water depletion followed by rehydration [6]. The capacity of polyploid trees to retain more water during dehydration than diploids has been observed in different plant genera including forest trees and shrubs (*Populus* [7], *Betula* [8], or *Lonicera* [9]) as well as domesticated tree crops (*Malus*, [10,11]; *Ziziphus*, [12]; *Citrus*, [13,14]), suggesting that polyploids might display an enhanced resilience to water shortage than diploids. Yet, field evaluations of the physiology of diploid and autotetraploid adult trees under drought conditions remain largely unexplored.

The stronger performance of polyploids under water deficit conditions has been correlated with differences in foliar elasticity and hygroscopicity between diploids and tetraploids, traits that correlate with cell size allometries, such as the escalation of stomatal guard cell size or the radius of vascular conduits [5]. Differences in cell size contrast with the conserved presence in the cell walls of pectins, versatile saccharidic molecules that play a role in foliar humectation, which further associate with tensile properties of cells, hypothesized to be the force of water pulling from the soil [15]. Specific pectins of the cell walls composing the phloem conduits strongly influence their functionality [16]. Examples include the branched pectin epitopes detected in sieve tube elements of the phloem within roots of herbaceous species [17], and in the secondary phloem of trees [18] and lianas [19]. Linear pectins have typically been associated with cells on the surface of leaves, such as trichomes or epidermal cells [20]. Despite the fact that pectins from the mesophyll participate in permeability to water and gases [21,22], cell-wall-related pectins in the foliar tissues have been poorly characterized in the leaves of most trees with exceptions such as *Capparis odoratissima* [23], and there are no reports comparing pectins between individuals of different ploidies.

In this work, we compared irrigated and non-irrigated diploid and autotetraploid mango leaves belonging to one tree per treatment of the cultivar ‘Kensington Pride’, in order to have a better knowledge of their chemical differences and preliminarily test the physiological performance of conspecific trees with doubled chromosome numbers under drought.

## 2. Results

### 2.1. Abiotic Variables

In the Mediterranean climate of the region in which the experiments were performed, precipitations in the winter season (November to February) of 2021 were above 120 mm, while average temperatures rarely dropped below 10 °C (Figure 1). However, these conditions became more extreme during the summer, when precipitations dropped to a minimum (i.e., close to zero from June to September), and average maximum temperatures reached values around 30 °C during the hottest months of July and August (Figure 1).

Volumetric water content in the soil with supplemental irrigation ranged between 70% of the maximum value all year long in the most superficial horizons (10–20 cm deep) and 80% in the deepest horizon layers (130–140 cm) (Figure 1). In contrast, non-irrigated soil plots displayed significantly lower water content even in the deepest horizons, which showed an average of 60% (*p* < 0.001), despite the fact that they were located in the same field. These differences became stronger at the superficial horizons (10–20 cm), with a volumetric water content of 30–40% during the summer season.

### 2.2. Fruit Production

Two years after the suppression of irrigation, the morphological differences between trees with and without irrigation were clear to the naked eye (Figure 2A–D). Irrigated trees from both cytotypes developed two new vegetative flushes during the growing season (the first one before flowering and the second after fruit set), which resulted in more branches and leaves (Figure 2A,C). In contrast, trees without supplemental irrigation suppressed the production of new flushes, and, during the hottest months of the summer, necrotic terminal branches (black-colored, without leaves) were commonly observed (Figure 2B,D). Inflorescence production was compromised in the diploid tree without irrigation (Figure 2A,B), but not in the tetraploid tree, which displayed similar inflorescence production than irrigated trees (Figure 2C,D). However, while an initial fruit set was observed in the non-irrigated tetraploid tree, many fruits aborted over the growing season. As a result, only a few fruits were obtained in the non-irrigated tetraploid tree. Despite the fact that fruits from the irrigated tetraploid tree were significantly heavier (F = 37.4, *p* < 0.05), wider (F = 20.8, *p* < 0.05), and longer (F = 5.3, *p* < 0.05) than fruits from the irrigated diploid (Figure 2E), brix degrees, which roughly measure sucrose concentration, were similar between fruits of both ploidies (F = 0.06, *p* < 0.05).

### 2.3. Leaf Water Potential and Stomatal Conductance

The midday leaf water potential of diploid leaves with irrigation on demand was maintained around −0.2 MPa or higher all year round. Irrigated tetraploid leaves showed similar values compared with diploids during the months with milder temperatures (i.e., up to initial fruit set), but during the hottest months, when most of the fruit development process takes place, their leaf water potential at midday dropped slightly to −0.3 MPa (Figure 3A). Strikingly, when irrigation was suspended, tetraploid leaves maintained their leaf water potential values around −0.3 MPa or higher even during the summer months with more evaporative demand, whereas leaf water potential in diploid leaves dropped to values close to −0.5 MPa, suggesting a differential response to soil water shortage between plants of both ploidies (Figure 3B). In fact, a two-way ANOVA at a *p* < 0.05 indicates that water potential variation was more attributed to ploidy than to irrigation.

Leaves from the diploid tree with supplemental irrigation reached the highest stomatal conductance measurements from June to September (that represents most of the fruit growing season) with values ranging between 500 and 700 mmol H_2_O m^2^ s^−1^, whereas leaves from tetraploids maintained twice lower values (values ranging between 350 and 500 mmol H_2_O m^2^ s^−1^) (Figure 3C). Leaves of trees without irrigation from both cytotypes reduced stomatal conductance (G_s_), but leaves of diploids maintained higher values (below 500 mmol H_2_O m^2^ s^−1^), compared with those of tetraploids (below 350 mmol H_2_O m^2^ s^−1^) (Figure 3D). Aggregated data revealed significant differences in G_s_ attributed to ploidy (F = 52.5, *p* < 0.001), irrigation (F = 27.4, *p* < 0.001), and their interaction (F = 5.83, *p* < 0.05). Comparing G_s_ by date of measurement, significant differences were attributed to ploidy in most months, but irrigation had a significant effect in August and September, when evapotranspiration demands were at their highest.

### 2.4. Cell Wall Biochemistry in Mango Leaves

The mango leaves accumulated high amounts of insoluble polysaccharides in the cell walls of the adaxial and abaxial epidermis and on the external side of vascular bundles (Figure 4A). The square cells of the adaxial epidermis contained very thick cell walls, contrasting with the thinner cell walls of elongated palisade parenchyma cells (Figure 4B). Tetraploid leaves showed conspicuously bigger cells than diploid leaves (Figure 4C). Polysaccharides accumulated in the external side of foliar vascular bundles, around the sclereids (Figure 4D), but also in phloem tissue in both ploidies (Figure 4E).

To obtain a deeper understanding of the nature of the saccharide compounds, the immunolocalization of two pectin epitopes in the cell walls revealed a clear topological pattern within leaves of both ploidies (note that Figure 5 displays only diploid leaves for simplicity). The LM5 pectin epitope populated the cell walls of the adaxial epidermis, the surrounding areas of vascular bundles, and the cell walls of the spongy mesophyll, but was missing in the abaxial epidermis (Figure 5C). Conversely, the LM26 branched epitope was not present in the adaxial epidermis, but abounded in the cell walls of the abaxial epidermis as well as in the chloroplast envelope (Figure 5D). The LM5 epitope was localized on the inner side of the cell wall in the adaxial epidermis and the tip of palisade parenchyma cells (Figure 5E), whereas the LM26 epitope displayed the profile of chloroplasts in the palisade parenchyma (Figure 5F). Spongy mesophyll cells contained a significant amount of the linear LM5 epitope (Figure 5G), but the membranes of abaxial epidermal cells specifically displayed the LM26 epitope (Figure 5H, see also [5]).

LM26 labeled the sieve tube elements of the vascular areas in leaves of both ploidies (Figure 6). The tertiary veins displayed an adaxial xylem and an abaxial phloem separated by a laticiferous canal and surrounded by a thick layer of fibers (Figure 6a,d). A comparison of leaves from both ploidies immunolocalized for the LM26 epitope revealed its presence in the sieve tube elements of the tertiary veins (Figure 6b,e), but also in the minor veins (Figure 6c,f). The differences in the size of the vascular elements that belong to comparable vein orders were obvious to the naked eye. (Figure 6a–f, but see [5] for quantitative data.)

## 3. Discussion

The extreme environmental conditions, with an insignificant amount of rainfall and high temperatures, of the summer season—which coincides with the mango fruit growing season in the southern Mediterranean coastal areas of Spain where mango is cultivated—reveal different physiological responses of diploid and autotetraploid ‘Kensington Pride’ mango leaves and inflorescences to soil water shortage. Despite the conservation of pectin moieties in the foliar cells of both cytotypes, increased cell sizes of tetraploid leaves may account for better hydration of foliar tissues.

### 3.1. Cell Wall Biochemistry in Leaves of Diploid and Tetraploid Mango Genotypes

The trade-off between cell turgidity and hydration crucially affects foliar tissues in a way that conditions the morphology–function relationship. Our work provides evidence of a similar cell wall biochemistry in leaves of both diploid and autotetraploid ‘Kensington Pride’ mango genotypes. The presence of polysaccharide-specific stains is associated with areas of high humectation within leaves, such as epidermal tissues or areas around vascular bundles. A finer detection of pectins revealed a chemical asymmetry between the adaxial and abaxial epidermises of the leaves. Pectins are universally associated with the hygroscopicity of cell walls, retaining water and thus contributing to key physiological aspects, such as the ascent of water through the xylem column or foliar water uptake [23,24]. Foliar water uptake may occur in both leaf surfaces, but it is more commonly observed in the abaxial side, where stomata constitute one of the gates for the entrance of water in some tree species (reviewed in [25]). While we did not analyze foliar water uptake in this work, we hereby provide evidence for the unique localization of the LM26 epitope in abaxial epidermal cells, suggesting that this epitope might be involved in the tensile properties of cell walls that typically adjust their volume in responses to environmental stimuli [5]. Another example of cells that undergo important hydrostatic pressures for their functionality are the sieve tube elements of the phloem, where this epitope is also present in the minor and major veins of diploid and tetraploid mango leaves, showing the ploidy-dependent size relationship of these conduits [size measured in reference [5]. The presence of this epitope in the sieve tube elements of the phloem was previously reported in vascular plants with different growth habits, such as roots of the herbaceous sugar beet [17], but also in the secondary phloem of trees such as poplar [18] or woody lianas such as *Austrobaileya* [19], suggesting their presence along the phloem across a wide range of taxa. Our work provides the first evidence of the presence of this moiety in the phloem of minor leaf veins in trees with doubled chromosome numbers.

Volumetric changes associated with the LM26 epitope affect not only cells but also organelles such as plastids. The consistent presence of this epitope in the plastids of diploid and tetraploid leaves, including in the palisade parenchyma and the spongy mesophyll, provides more evidence of their involvement in organelles that undergo relatively rapid volumetric changes, as is the case of diurnal starch accumulation within plastids. The capacity of plastids to assimilate CO_2_ has been associated with their surface-to-leaf area [26]. While the possible localization of this epitope in the plastids requires confirmation in other species, it could represent a promising approach to quantifying the anatomical limitations associated with CO_2_ distribution within leaves. 

Our work further reveals the concomitant localization of insoluble polysaccharides in foliar tissues with a high hygroscopicity potential, such as the areas surrounding vascular bundles, which participate in the exchange of water between conductive and parenchymatous tissues. The strong presence of the LM5 pectin epitope in the cell walls of the spongy mesophyll and the vascular bundles of both diploid and tetraploid ‘Kensington Pride’ mango leaves suggests an involvement of pectins in diffusivity of compounds dissolved in water within leaves. Previous research associated the presence of pectins in the mesophyll of different species [27,28,29] with CO_2_ conductance in the mesophyll [21], further suggesting their involvement in the adaptation of plants to water shortages [22]. Pectins of the cell walls offer an extreme versatility that may contribute significantly to the resilience of plants to abiotic stresses, but the effect of ploidy on the specific patterning of pectins within foliar tissues is beginning to be elucidated [30] and requires a combination of studies on the anatomy and functional performance of plants under stress in field conditions.

### 3.2. Tetraploid Leaves of ‘Kensington Pride’ Mango Display a Higher Efficiency Than Diploids under Soil Water Deficit

In this work, we qualitatively confirmed the gigas effect (the increased size of polyploid plant cells derived from multiplied chromosome content that results in enlarged plant fruit organs (see [31]) in autopolyploids of the mango cultivar ‘Kensington Pride’. Enlarged cells of polyploids include not only the vascular cells and the stomata, as previously reported [5], but also the aerial organs (leaves, flowers and fruits). Additionally, the escalation of size affects the cells of different tetraploid foliar tissues, including the epidermis and parenchymatic tissues. An increase in fruit size was previously observed in other subtropical fruit tree polyploids, such as genotypes of the pantropical genus *Annona* [31]. However, the results obtained in this work show that the percentage of soluble sugars accumulated in mango fruits is similar between ploidies. This maintenance of sugar concentrations in the sink tissues of the larger tetraploid fruits calls for the possibility of a higher carbon fixation rate in polyploids, although additional tests are under development. Yet, autotetraploid leaves consistently display lower stomatal conductance than diploids in the field under good irrigation, which is associated with their larger and less dense stomata [5]. The size–density trade-off reflected in lower stomatal conductance has previously been observed in other plant species, both in herbaceous grasses [32] and perennial trees [33,34] and has been associated with higher water use efficiency in polyploids. The question remains whether lower conductance could be a predictor of drought tolerance, as reported for crops with lower stomatal densities [35,36,37], although some triploid trees have been shown to have increased stomatal conductance yet reduced internal water use efficiency [8,38,39]. Due to the low availability of polyploid genotypes of mango, in this work, we subjected one mango tree from each ploidy to scenarios of water deficit, suppressing supplemental irrigation for two years in a Mediterranean climate with extremely dry summers. Despite the fact that both treatments were separated by a distance of just 3 m, under no supplemental irrigation, foliar stomatal conductance was reduced in both diploid and tetraploid ‘Kensington Pride’ mango genotypes, but this reduction was more dramatic in diploid leaves, suggesting their higher susceptibility to localized lack of water in the soil compared with tetraploids, in line with previous reports on different tetraploid varieties of *Citrus* rootstocks [40,41]. Some previous reports have also suggested a higher drought resistance of polyploid tree species [34,42]. The possible coupling between stomatal behavior and hydraulic function [43], measured as leaf water potential, revealed high water potential in mango regardless of the water content in the soil. Interestingly, tetraploid leaves had slightly lower potential under good irrigation and, unlike the more variable potential of diploid leaves, tetraploids maintained their values under soil water reduction. This maintenance could be attributed to the isohydric character or some mango varieties such as ‘Kensington Pride’ [5].

At the whole-plant level, soil water shortage impaired new biomass production in both diploid and tetraploid genotypes, pointing to irrigation as crucial for good productivity of this crop under our climatic conditions. Indeed, rainfall dropped sharply from 60.5 mm on average in May to 5 mm in June, being almost 0 in July and August, when maximum temperatures reached around 38 °C. This lower biomass production is associated with the suppression of reproductive development in the following spring, which was more dramatic in the diploid than in the tetraploid tree. Although we are aware of the limitations of our study in terms of number of trees analyzed, the preliminary results obtained show that the processes of flower induction and flower development are less affected by soil water deficit in tetraploids, while supplemental irrigation is required for appropriate fruit set and development. These preliminary results open the way for further studies in the following years when additional trees are available and a study at a larger scale can be designed.

## 4. Materials and Methods

### 4.1. Plant Material

Autotetraploid mango genotypes are rare, but among the four trees of mango cv ‘Kensington Pride’ maintained in our mango germplasm collection at IHSM La Mayora (cX:407.162,62; Y:4.068.652,56; UTM:30), through a combination of molecular markers and flow cytometry (the combination of both methods avoids confusion with possible aneuploids or mixoploids), two of them were characterized as diploids and the other two as autotetraploids (published in [5]). Due to the limited number of trees available, and the preliminary character of this study, our experimental design considered the leaves, not the trees, as the unit sample. The trees were planted in two parallel rows, one for each ploidy, separated by 3 m. In order to analyze the effect of soil water stress, one tree from each ploidy was left with a regular irrigation schedule (during the growing season, three times per week two hours each, totaling a volume of 48 L per week), and one tree per ploidy had irrigation suppressed in September 2019. Trees were left to acclimate to these conditions for two years, and phenological and physiological changes were qualitatively observed during the flowering and fruit growing seasons. Mature fruits were collected from the four trees for two years to measure their size (length and width) with a caliper, their weight with a fine scale, and degrees Brix, which provide a general idea of the sugars dissolved per volume unit, with a refractometer.

### 4.2. Environmental Variables and Physiological Measurements

To monitor the humidity of the soil, we used FDR capacitance sensors (TBagrosensor, Valencia, Spain) that track the humidity of the soil for each irrigation treatment at different horizons belowground each separated by 10 cm, from the surface to a 140 cm depth. The sensors were calibrated at installation by saturating the contacting soil area with water, according to the manufacturer’s instructions. Measures were taken every 15 min, sent through a wireless system to a server, and recorded. Rainfall, relative humidity, and minimum and maximum temperatures were also tracked and recorded with environmental sensors.

Midday leaf water potential was evaluated bi-weekly from April to September using a Scholander pressure chamber (model 600; PMS Instrument, Albany, OR, USA), https://www.pmsinstrument.com, accessed on 1 September 2019) in 3 fully developed mature leaves per tree (*n* = 12 in all treatments), avoiding leaves from new growth flushes, which typically display water potentials close to 0 (according to preliminary trials). Concomitantly, stomatal conductance (G_s_, mmol H_2_O m^−2^ s^−1^) was estimated with a steady-state leaf porometer (SC-1, Decagon Devices, USA) on four fully developed leaves per tree under the two treatments (*n* = 16 in all treatments) oriented to the four cardinal points. The sensor head was calibrated daily before performing G_s_ measurements in the field according to the manufacturer’s protocol.

### 4.3. Histochemistry and Immunolocalization of Pectin Polysaccharides

In order to study the polysaccharides present in the leaf tissues, the tertiary veins and the aerials of three leaves per ploidy (*n* = 3) were fixed in 100% dry methanol, then washed in 100% ethanol three times for 1 h each [44] prior to embedding in Technovit8100 methacrylate resin under anoxic conditions. Blocks were sectioned at 4 µm with a Leica EM UC7 ultramicrotome (Leica Microsystems, Wetzlar, Germany) and mounted onto slides coated with 3-Aminopropyl triethoxysilane (Sigma Aldrich, St. Louis, MI, USA) for immunolocalization. Polysaccharides were stained in pink with a periodic acid Shiffs reagent-PAS and counterstained with 0.025% aqueous toluidine blue that stains for general tissue structure in blue. Two monoclonal antibodies (mAb) were used, LM5 (Plant Probes, Leeds, UK, http://www.plantprobes.net, accessed on 20 October 2019), which recognizes a linear pectic 1,4-galactan epitope in the cell walls, and LM26 (Plant Probes, Leeds, UK), which detects a branched β-1,6-galactosyl substitution of pectic β-1,4-galactan, a pectic rhamnogalacturonan-I [17]. Negative controls were treated in the same way but substituting the primary antibody with a solution of 1% bovine serum albumin in 1× phosphate buffer saline (PBS).

### 4.4. Statistical Analysis

Soil water content was averaged per month and compared across similar horizons between both irrigation treatments using a one-way ANOVA at a *p* < 0.05. Fruit weight, length, width, and degrees Brix were compared between ploidies using a one-way ANOVA at a *p* < 0.05. Aggregated data of G_s_ and Ψ were compared with a two-way ANOVA using ploidy and irrigation as independent variables. The same test was used to evaluate each day of measurement individually.

## 5. Conclusions

Our work is based on preliminary data obtained from few samples, but suggests a better performance of autotetraploid mango trees under soil water deficit scenarios, supporting the long-held hypothesis that polyploids have more robust performance under water stress than diploids [45,46]. Work is underway to plant a new trial involving more polyploid mango genotypes (see [6]) and, once the trees reach full production, those results will serve to confirm our observations with other polyploid tree species [34]. Altogether, these studies could result in more sustainable orchard management under the current situation of climate-change-induced reduced rainfall in different areas of the world.

## Figures and Tables

**Figure 1 plants-12-03738-f001:**
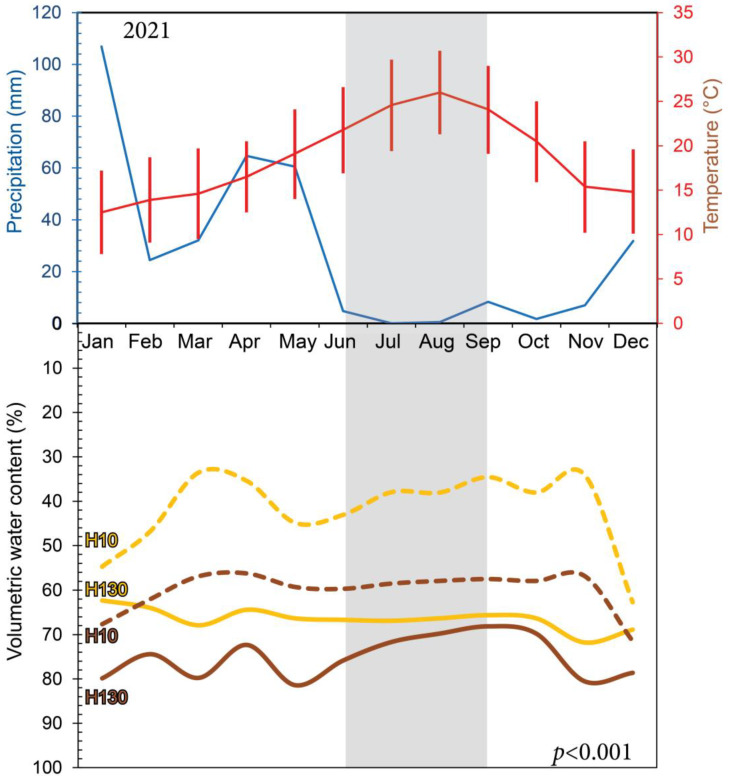
Precipitation aboveground (**top**) and volumetric water content belowground (**bottom**) during 2021. Mean monthly precipitation (blue color) was minimal during the summer months, whereas mean monthly temperatures (red color) reached maximum values. Bars represent the maximum and minimum average temperature for each month. Volumetric water content was minimal in non-irrigated soils (orange), especially in the superficial horizon of 10 cm (dotted lines, H10). In contrast, deeper horizons (continuous lines) maintained higher percentages of water per soil volume, especially in irrigated areas of the soil. *p*-Value represents significant differences of the relative water in the soil across comparable horizons. Shaded area corresponds with the period when measurements were taken in this work.

**Figure 2 plants-12-03738-f002:**
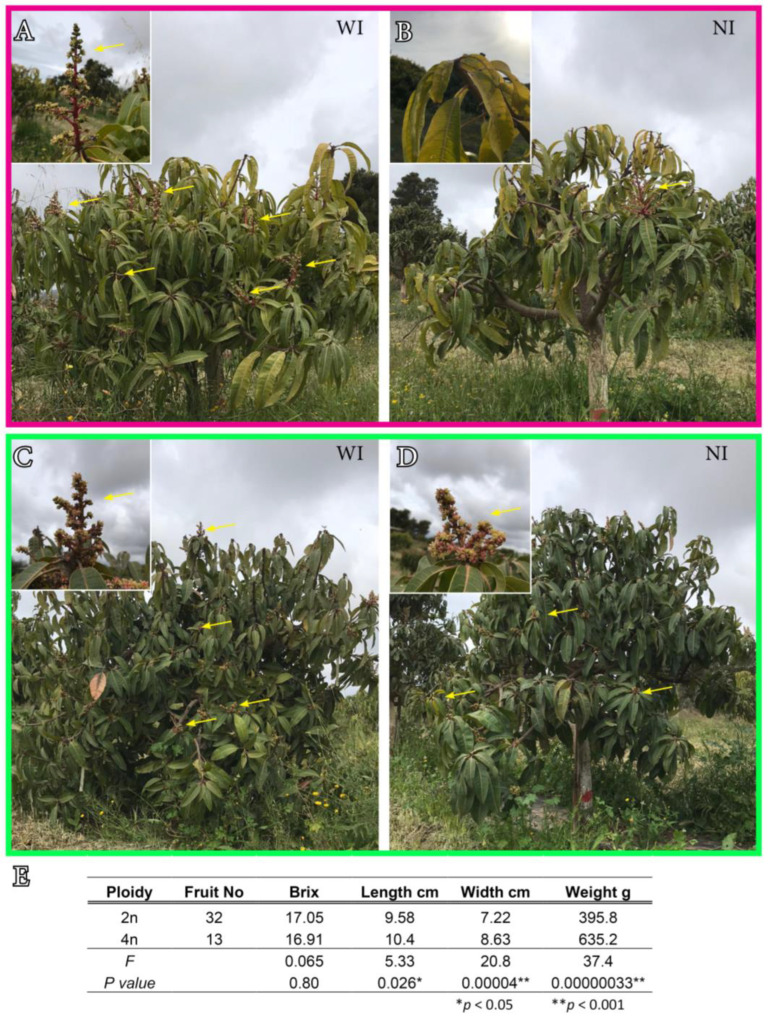
Morphology of ‘Kensington Pride’ mango trees under well-irrigated (WI) and non-irrigated conditions (NI) comparing diploid (pink) and tetraploid (green) trees during the spring of 2021. (**A**). Well-irrigated diploid mango tree with numerous inflorescences (inset). (**B**). Non-irrigated diploid mango tree had lower biomass and a few inflorescences (inset). (**C**). Well-irrigated tetraploid mango tree showing a dense canopy and numerous inflorescences (inset). (**D**). Non-irrigated tetraploid mango tree with less biomass, but several inflorescences. (**E**). Morphological and chemical measurements of fruits from diploid and tetraploid genotypes under normal irrigation. Yellow arrows point to inflorescences.

**Figure 3 plants-12-03738-f003:**
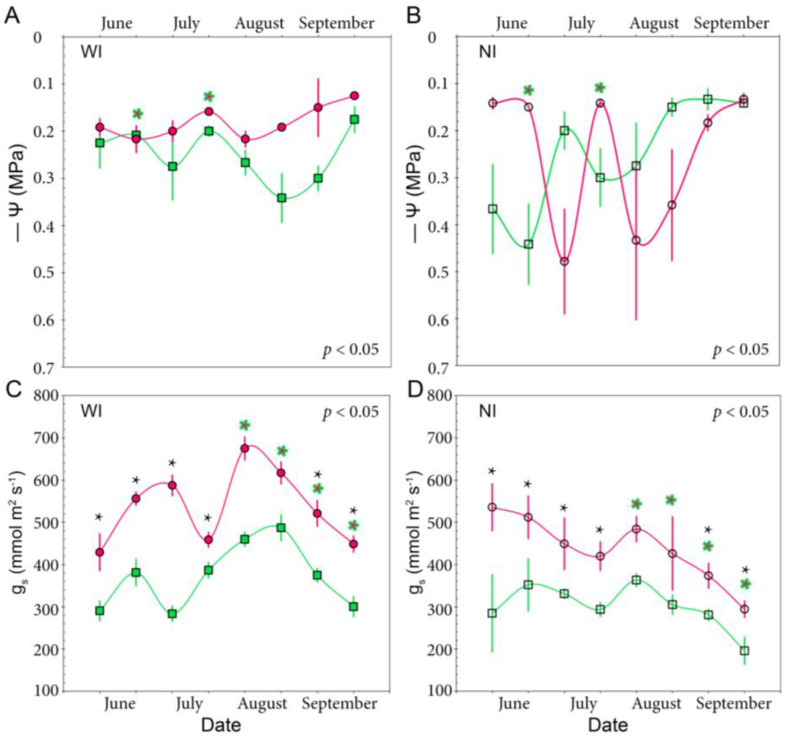
Leaf water potential at midday (Ψ) and stomatal conductance (G_s_) in diploid (pink) and tetraploid (green) ‘Kensington Pride’ mango trees. (**A**). Ψ in well-irrigated (WI) diploid and tetraploid mango trees. (**B**). Ψ in diploid and tetraploid mango trees without supplemental irrigation (NI). (**C**). G_s_ in well-irrigated diploid and tetraploid mango leaves. (**D**). G_s_ in diploid and tetraploid mango leaves without supplemental irrigation. Bars represent standard error at a *p* < 0.05, and stars represent significant differences attributed to ploidy (colored stars) or irrigation (black stars) after a two-way ANOVA at a *p* < 0.05.

**Figure 4 plants-12-03738-f004:**
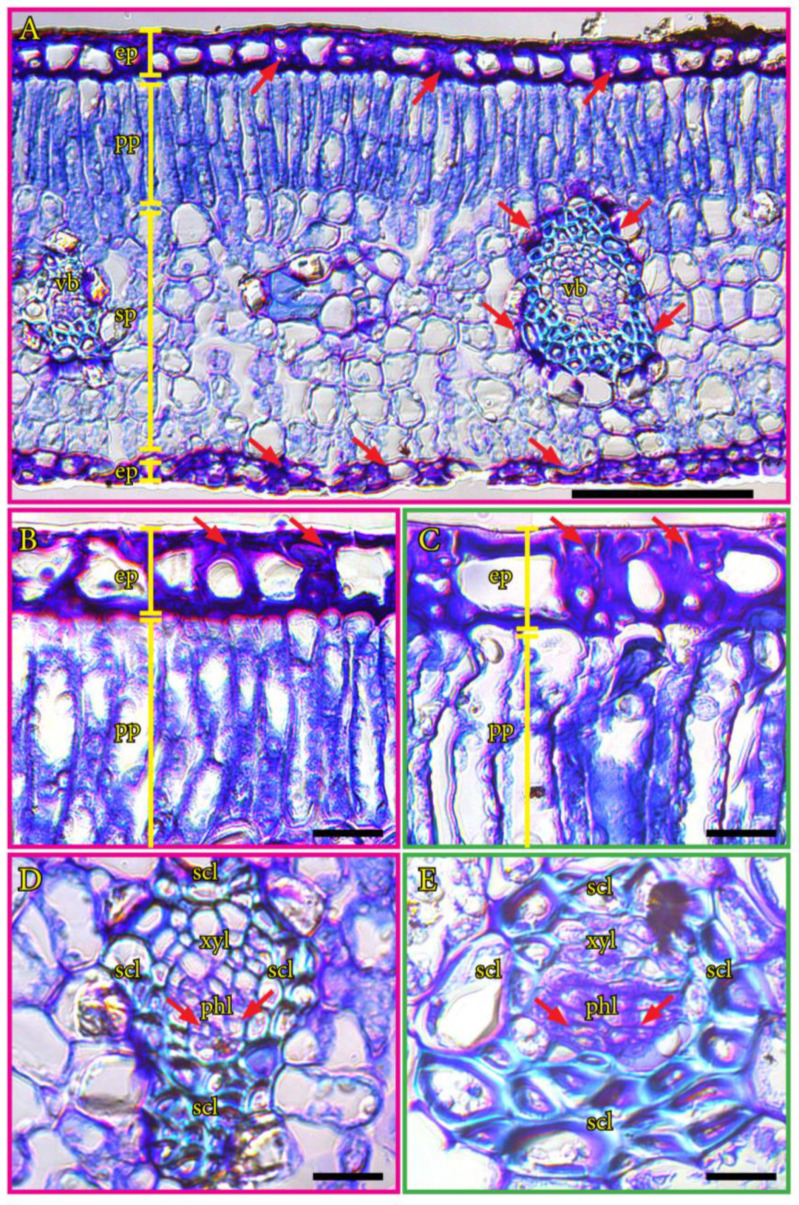
Histochemistry of leaves from diploid (pink) and tetraploid (green) ‘Kensington Pride’ mango genotypes. (**A**). Cross section of a leaf from a diploid mango genotype showing the structure of the different layers and the accumulation of insoluble polysaccharides (red arrows). (**B**). Detail of the adaxial side of a leaf from a diploid mango genotype with a strong accumulation of insoluble polysaccharides in epidermal tissue (red arrows). (**C**). Detail of the adaxial side of a tetraploid leaf with larger cells and cell walls showing stained polysaccharides (red arrows). (**D**). Cross section of a minor vein in a leaf from a diploid mango genotype showing conductive tissues. (**E**). Cross section of a minor vein in a leaf from a tetraploid mango genotype showing the accumulation of polysaccharides in the walls of phloem cells (red arrows). Sections stained with periodic acid Shiffs-PAS for insoluble polysaccharides (pink) and counterstained with 0.025% toluidine blue for general structure (blue-purple). Ep, epidermis; phl, phloem; pp, palisade parenchyma; scl, sclerenchyma; vb, vascular bundle; xyl, xylem. (**A**) Scale bar: 100 µm; (**B**–**E**) scale bars: 20 µm.

**Figure 5 plants-12-03738-f005:**
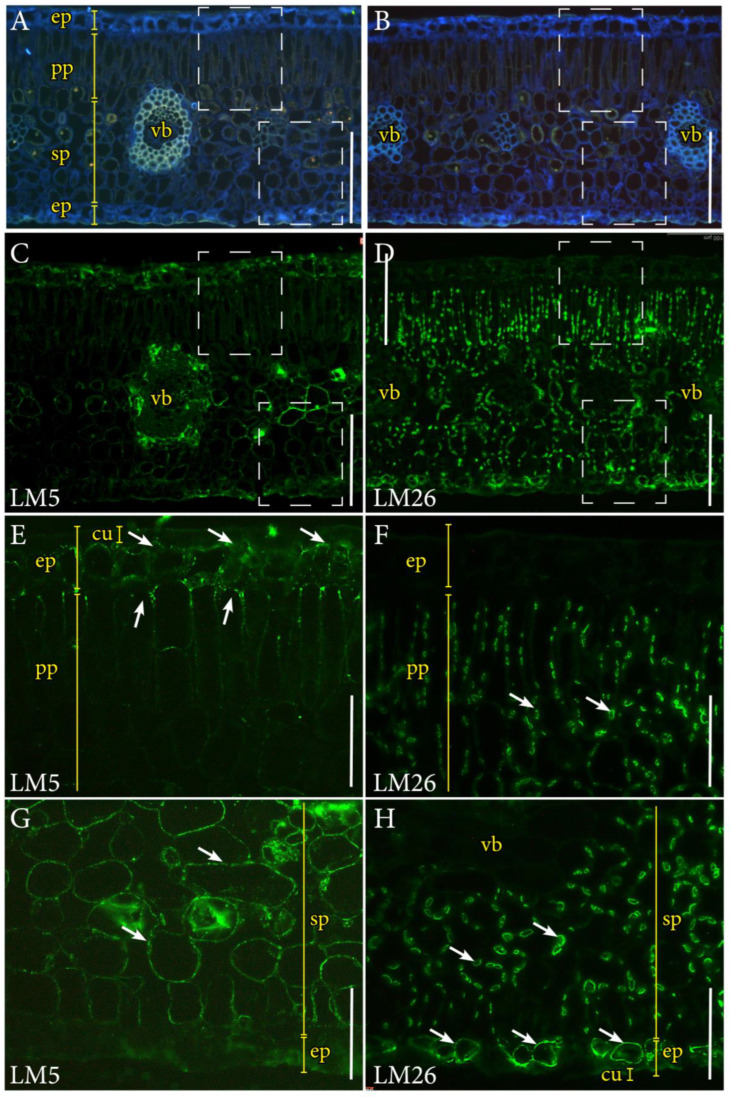
Immunolocalization of pectin epitopes with the monoclonal antibodies (mAbs) LM5 and LM26 in ‘Kensington Pride’ mango leaf tissues. (**A**,**B**). Cross sections of leaves from a diploid mango genotype showing the different tissues under UV autofluorescence. (**C**). Same section as in (**A**) immunolocalized with the LM5 mAb. (**D**). Same section as (**B**) immunolocalized with the LM26 mAb. (**E**). Detail of the adaxial side of the leaf (top dotted square in (**A**) and (**C**)) displaying the presence of the LM5 epitope in the cell walls of the adaxial epidermis and the tip of palisade parenchyma cells (arrows). (**F**). Details of the adaxial epidermis (top dotted square in (**B**) and (**D**)) showing an absence of the LM26 epitope but a strong presence in the plastids (arrows). (**G**). Detail of the abaxial side of the leaf (bottom dotted square in (**A**) and (**C**)), showing the presence of the LM5 epitope in the cell walls of the spongy mesophyll (arrows). (**H**). Detail of the abaxial side of the leaf (bottom dotted square in (**B**) and (**D**)) showing the presence of the LM26 epitope in the plastids of the spongy mesophyll and the abaxial epidermis (arrows). (**C**–**H**) Fluorescent green signal of the Alexa488 secondary antibody attached to each primary mAb. Cu, cuticle; ep, epidermis; pp, palisade parenchyma; sp, spongy parenchyma; vb, vascular bundle. (**A**–**D**) Scale bars: 100 µm; (**E**–**H**) scale bars: 50 µm.

**Figure 6 plants-12-03738-f006:**
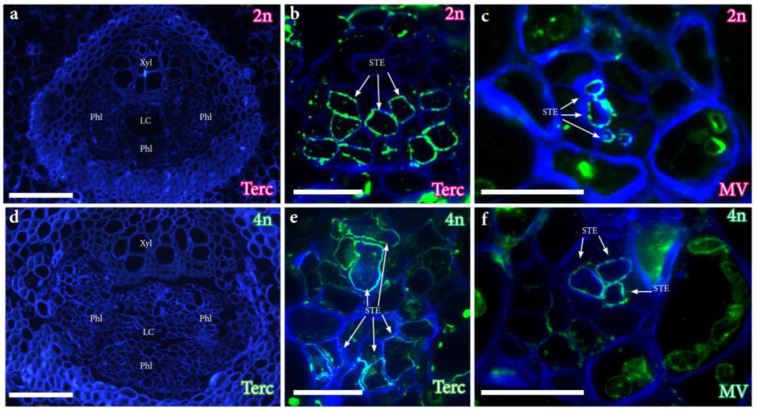
Immunolocalization of the LM26 pectin epitope in the vascular tissues of leaves from diploid and tetraploid ‘Kensington Pride’ mango genotypes. (**a**). Tertiary vein of a leaf from a diploid genotype in cross section displaying the localization of the xylem (Xyl), the phloem (Phl), and the laticiferous canal (lc). (**b**). Detail of phloem tissue, displaying the localization of the LM26 epitope in the sieve tube elements (arrows). (**c**). Minor veins showing the specific localization of the sieve tube elements with LM26 mAb. (**d**). Tertiary vein of a leaf from a tetraploid genotype in cross section. (**e**). Detail of the phloem in a leaf from a tetraploid genotype showing the localization of the LM26 epitope in the sieve tube elements. (**f**). Minor veins of the leaf of a tetraploid genotype showing the sieve tube elements stained with the LM26 epitope. (**a**,**d**) Scale bars: 100 µm; (**b**,**c**,**e**,**f**) scale bars: 20 µm.

## Data Availability

Data generated in this work are available upon request to the corresponding author.

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
