# Peer review of "Foliar Pectins and Physiology of Diploid and Autotetraploid Mango Genotypes under Water Stress"

_plants, 2023, doi:10.3390/plants12213738_

Round 1

Reviewer 1 Report

Comments and Suggestions for Authors

1.      The novelty of this work is not clear comparing with existing works discussed in Section 1.

2.      The main contributions of this manuscript are not summarized clearly in the manuscript. The authors should also add discussion about differences between the proposed work and related works.

3.      The contents in this manuscript are organized poorly. There are lots of redundant irrelevant details in the manuscript.

4.      There are a lot of grammar mistakes, typing errors in the manuscript.

5.      The labels and the references of figures are inconsistent throughout the manuscript.

6.      Normaly Material and Methods is section 2. but you added it after results and it is not good. please replace.

Author Response

  1.     The novelty of this work is not clear comparing with existing works discussed in Section 1.

We kindly disagree with the reviewer, and we have stressed that this is the first preliminary trial evaluation in the field comparing adult diploid and tetraploid mango trees.

  1.     The main contributions of this manuscript are not summarized clearly in the manuscript. The authors should also add discussion about differences between the proposed work and related works.

We have revised the work following the suggestion of the reviewer but related works are very scarce, as stated previously.

  1.     The contents in this manuscript are organized poorly. There are lots of redundant irrelevant details in the manuscript.

We have revised the work trying to follow the suggestion of the reviewer, although additional details would have been desirable in order to know exactly what the reviewer thinks that is redundant/irrelevant.

  1.     There are a lot of grammar mistakes, typing errors in the manuscript.

Thanks for the comment. We have revised the manuscript thoroughly.

  1.     The labels and the references of figures are inconsistent throughout the manuscript.

We have rechecked all the figure legends for clarity.

  1.     Normaly Material and Methods is section 2. but you added it after results and it is not good. please replace.

We have followed the requirements of the format of the journal.

We appreciate the comments of the reviewer but it is difficult to address the criticisms since not enough information is provided in the review.

Reviewer 2 Report

Comments and Suggestions for Authors

Fonollá et al. present an interesting study about the physiological performance of diploid and autotetraploid mango trees under drought conditions. The objective is novel for this crop and the need for and importance of the study is well justified. Furthermore, the data are correctly presented, and the manuscript is brief and easy to read.

Major comments:

-        Experimental design: only one plant (n=1) per treatment and ploidy level is considered. Instead, the authors “considered the leaves, not the trees, as the unit sample” (L342-343). This approximation is incorrect, since the replicates are not statistically independent, which very significantly limits the possible conclusions of this study. Although this point cannot be resolved without repeating the experiment with a larger number of individuals, if this manuscript is published it is essential that this limitation be made clear both in the abstract and at the end of the introduction in order to that all readers take it into account.

-        Tree planting framework. I consider that a distance between trees of 3 meters is insufficient to adequately define the different irrigation treatments, since in adult trees the roots can reach beyond this distance. This would explain the lack of differences in soil moisture in the deeper horizons and in leaf water potential at midday. Without a doubt, drought treatment has effects on the physiology of these trees, but the experimental design is difficult to replicate.

-        Leaf anatomy. Although the location of the different types of pectins can only be qualitative, any other mention of anatomical parameters (e.g. cell size, cell wall thickness, etc.) must be measured quantitatively and data presented in table format with the relevant statistics.

Finally, I raise several minor comments below.

Comments by line:

L14 – Please revise this sentence “Polyploids are alternative genotypes with potential water use efficiency”. Are the authors referring to greater water use efficiency?

L16-18 - Specify that these are plants planted in the ground and the period in which the different evaluations were performed.

L20 – This sentence is confusing: “Autotetraploids displayed lower Ψl and Gs, but the lack of irrigation only affected Gs.” Does the first part of the sentence refer to well-watered conditions?

L23 – “fruit organ size”?

L24 – “Specific cell wall hygroscopic materials” refer to pectins?

L44 – Remove “more”.

L47 - I think it is worth further explaining the findings reported in reference 6, since the phrase where it is currently cited is very general.

L47 – What does “maintain more water” mean?

L73 - If there is only one tree per treatment and genotype, it is not possible to refer to “non-irrigated tetraploid trees”, “Irrigated tetraploid trees”, “diploid trees without irrigation”, etc. Please revise it along the entire manuscript.

L142 – If stomatal conductance is measured with a leaf porometer, which measures diffusion conductance by comparing the rate of humidification (i.e. water) within the chamber to readings obtained with a calibration plate, how did the authors calculate the stomatal conductance to CO2 diffusion?

Author Response

REVIEWER 2

Fonollá et al. present an interesting study about the physiological performance of diploid and autotetraploid mango trees under drought conditions. The objective is novel for this crop and the need for and importance of the study is well justified. Furthermore, the data are correctly presented, and the manuscript is brief and easy to read.

We very much appreciate the positive detailed feedback of the reviewer.

Major comments:

-        Experimental design: only one plant (n=1) per treatment and ploidy level is considered. Instead, the authors “considered the leaves, not the trees, as the unit sample” (L342-343). This approximation is incorrect, since the replicates are not statistically independent, which very significantly limits the possible conclusions of this study. Although this point cannot be resolved without repeating the experiment with a larger number of individuals, if this manuscript is published it is essential that this limitation be made clear both in the abstract and at the end of the introduction in order to that all readers take it into account.

We appreciate the comment, and have made this point clear in the abstract (L17-18), at the end of the introduction (L70), and at the end of the discussion (L397-399), as suggested.

-        Tree planting framework. I consider that a distance between trees of 3 meters is insufficient to adequately define the different irrigation treatments, since in adult trees the roots can reach beyond this distance. This would explain the lack of differences in soil moisture in the deeper horizons and in leaf water potential at midday. Without a doubt, drought treatment has effects on the physiology of these trees, but the experimental design is difficult to replicate.

We kindly disagree with this opinion. Actually, we observe significant differences in the soil water content of superficial horizons that are translated in marked different stomatal conductances. The similar leaf water potential might be attributed to an isohydric behaviour of this variety, as pointed out in the reference [6]. We pointed this out in L335-336.

-        Leaf anatomy. Although the location of the different types of pectins can only be qualitative, any other mention of anatomical parameters (e.g. cell size, cell wall thickness, etc.) must be measured quantitatively and data presented in table format with the relevant statistics.

Finally, I raise several minor comments below.

We previously quantified cell size [5], comparing diploids and tetraploids. For the purposes of the current work, we believe that a qualitative description is sufficient.

Comments by line:

L14 – Please revise this sentence “Polyploids are alternative genotypes with potential water use efficiency”. Are the authors referring to greater water use efficiency?

Thank you for the note. This has been modified (L14).

L16-18 - Specify that these are plants planted in the ground and the period in which the different evaluations were performed.

This was clarified, thanks (L16-18).

L20 – This sentence is confusing: “Autotetraploids displayed lower Ψl and Gs, but the lack of irrigation only affected Gs.” Does the first part of the sentence refer to well-watered conditions?

This sentence has been clarified (L19).

L23 – “fruit organ size”?

Yes, this has been added (L22).

L24 – “Specific cell wall hygroscopic materials” refer to pectins?

Yes, this has been clarified (L24).

L44 – Remove “more”.

More has been removed (L43-44).

L47 - I think it is worth further explaining the findings reported in reference 6, since the phrase where it is currently cited is very general.

More details of these findings are provided (L45-46).

L47 – What does “maintain more water” mean?

This has been clarified (L59).

L73 - If there is only one tree per treatment and genotype, it is not possible to refer to “non-irrigated tetraploid trees”, “Irrigated tetraploid trees”, “diploid trees without irrigation”, etc. Please revise it along the entire manuscript.

We have replaced trees by either `one tree´or `leaves´ along the manuscript.

L142 – If stomatal conductance is measured with a leaf porometer, which measures diffusion conductance by comparing the rate of humidification (i.e. water) within the chamber to readings obtained with a calibration plate, how did the authors calculate the stomatal conductance to CO2 diffusion?

Although this information is provided directly by the porometer, which is calibrated in each measurement according to manufacturer instructions (L382-385), stomatal conductance is estimated as the reciprocal of the resistance (converted to moles to avoid the effect of pressure) through which the concentration difference of water vapour fluxes. The formula are available elsewhere.
